# Exploring the Interplay Between Gut Microbiota and the Melatonergic Pathway in Hormone Receptor-Positive Breast Cancer

**DOI:** 10.3390/ijms26146801

**Published:** 2025-07-16

**Authors:** Aurora Laborda-Illanes, Soukaina Boutriq, Lucía Aranega-Martín, Daniel Castellano-Castillo, Lidia Sánchez-Alcoholado, Isaac Plaza-Andrades, Jesús Peralta-Linero, Emilio Alba, José Carlos Fernández-García, Alicia González-González, María Isabel Queipo-Ortuño

**Affiliations:** 1Medical Oncology Clinical Management Unit, Virgen de la Victoria University Hospital, Málaga Biomedical Research Institute (IBIMA) BIONAND Platform-CIMES-UMA, 29010 Malaga, Spain; aurora.laborda@ibima.eu (A.L.-I.); soukaina@ibima.eu (S.B.); lucia.aramar97@uma.es (L.A.-M.); daniecas22@gmail.com (D.C.-C.); l.sanchez.alcoholado@ibima.eu (L.S.-A.); isaac.plaza.andrades@ibima.eu (I.P.-A.); jesus.peralta@ibima.eu (J.P.-L.); ealbac@uma.es (E.A.); iqueipo@uma.es (M.I.Q.-O.); 2Department of Medicine and Pediatrics, Faculty of Medicine, University of Malaga, 29071 Malaga, Spain; 3Department of Endocrinology and Nutrition, Regional University Hospital of Malaga, Biomedical Research Institute of Malaga (IBIMA), Faculty of Medicine, University of Malaga, 29010 Malaga, Spain; jcfernandez@uma.es; 4Department of Surgical Specialties, Biochemical and Immunology, Faculty of Medicine, University of Málaga, 29071 Malaga, Spain

**Keywords:** serotonin-NAS-melatonin axis, gut microbiota, hormone receptor-positive breast cancer, dysbiosis, melatonergic pathway, permeability, SCFAs, beta-glucuronidase activity

## Abstract

Emerging evidence suggests a bidirectional relationship between gut microbiota, melatonin synthesis, and breast cancer (BC) development in hormone receptor-positive patients (HR+HER2+ and HR+HER2-). This study investigated alterations in gut microbiota composition, the serum serotonin–N-acetylserotonin (NAS)–melatonin axis, fecal short-chain fatty acids (SCFAs) and beta-glucuronidase (βGD) activity, and serum zonulin in HR+ BC patients compared to healthy controls. Blood and fecal samples were analyzed using mass spectrometry for serotonin, NAS, melatonin, and SCFAs; ELISA for AANAT, ASMT, 14-3-3 protein, and zonulin; fluorometric assay for βGD activity; and 16S rRNA sequencing for gut microbiota composition. HR+ BC patients exhibited gut dysbiosis with reduced *Bifidobacterium longum* and increased *Bacteroides eggerthii*, alongside elevated fecal βGD activity, SCFA levels (e.g., isovaleric acid), and serum zonulin, indicating increased intestinal permeability. Serum serotonin and N-acetylserotonin (NAS) levels were elevated, while melatonin levels were reduced, with a higher NAS/melatonin ratio in BC patients. AANAT levels were increased, and ASMT levels were decreased, suggesting disrupted melatonin synthesis. *Bifidobacterium longum* positively correlated with melatonin and negatively with βGD activity, while *Bacteroides eggerthii* showed a positive correlation with βGD activity. These findings suggested that gut microbiota alterations, disrupted melatonin synthesis, microbial metabolism, and intestinal permeability may contribute to BC pathophysiology. The NAS/melatonin ratio could represent a potential biomarker, necessitating further mechanistic studies to confirm causality and explore therapeutic interventions.

## 1. Introduction

Breast cancer (BC) remains a major global health burden, with hormone receptor-positive subtypes (HR+HER2+ and HR+HER2-) posing particular challenges in diagnosis and treatment. While traditional risk factors such as prolonged hormonal exposure are well documented, emerging evidence underscores the roles of circadian disruption and gut dysbiosis in BC pathophysiology. Notably, the gut microbiota has gained attention as a critical modulator of hormonal balance and cancer risk, largely through its influence on estrogen metabolism.

Bacteria expressing beta-glucuronidase (βGD) can deconjugate estrogens in the gut, promoting their reabsorption and increasing systemic estrogen levels, a well-established driver of hormone-related BC. In contrast, melatonin, a neurohormone with antiestrogenic, anti-inflammatory, and oncostatic properties, counteracts these effects by modulating enzymes involved in estrogen metabolism [1,2,3]. Importantly, melatonin is synthesized not only in the pineal gland but also in large amounts in the gastrointestinal tract, where it interacts closely with the gut microbiota [4,5,6]. The gut microbiota can influence melatonin production both directly and indirectly, by stimulating serotonin synthesis, a key precursor in the melatonergic pathway [5,6].

Recent studies highlight a bidirectional relationship between melatonin, synthesized via the tryptophan–serotonin–N-acetylserotonin (NAS)–melatonin pathway (involving arylalkylamine N-acetyltransferase (AANAT) and acetylserotonin-O-methyltransferase (ASMT) enzymes [7,8,9]), and gut microbial composition, with dysregulation linked to breast cancer (BC) development through altered tryptophan metabolism toward the kynurenine pathway [10,11]. Melatonin promotes gut homeostasis by increasing beneficial bacteria like *Lactobacillus crispatus*, *Bifidobacterium*, and *Akkermansia muciniphila*, while reducing pathogenic taxa such as Clostridiales, *Escherichia coli*, *Helicobacter pylori*, *Bacteroides massiliensis*, and Erysipelotrichaceae; however, BC patients with poor sleep quality show higher Firmicutes and pro-inflammatory *Ruminococcus* with βGD activity, suggesting reduced melatonin synthesis due to circadian disruption [12,13,14,15,16].

In parallel, short-chain fatty acids (SCFAs) such as butyrate—produced by microbial fermentation—support melatonin synthesis and preserve intestinal barrier integrity. Dysbiosis, frequently observed in BC patients, can reduce SCFA levels, alter tryptophan metabolism, increase the NAS/melatonin ratio [10,17], and contribute to intestinal inflammation and barrier dysfunction [13]. These factors may collectively promote tumor initiation and progression.

Despite these insights, the combined role of the gut microbiota and the melatonergic pathway in hormone receptor-positive BC remains poorly characterized. Few studies have simultaneously assessed microbial composition, melatonin biosynthesis (including serotonin and NAS), βGD activity, SCFAs levels, and intestinal permeability markers such as zonulin.

This study aimed to address this gap by investigating the interplay between gut microbiota alterations and dysregulation of the melatonergic pathway in HR+ BC patients. Specifically, we evaluated microbiota composition; serum levels of serotonin, NAS, melatonin, AANAT, ASMT, and zonulin; as well as fecal levels of SCFAs and βGD activity.. We also explored the potential of the NAS/melatonin ratio as a diagnostic biomarker. Our central hypothesis was that disturbances in the melatonergic pathway and gut microbial metabolism are interrelated processes that synergistically contribute to BC pathogenesis.

## 2. Results

### 2.1. Clinical Characteristics of the Study Groups

Table 1 presents the clinicopathological characteristics of the study groups. This study compared HR+ BC patients and healthy controls, with no significant differences in age (*p* = 0.72) or menopausal status (mostly postmenopausal, *p* = 1.00), minimizing confounding effects on microbiota and metabolic outcomes. Among 76 BC patients, 80.26% had invasive ductal carcinoma, 42.11% had tumors <2 cm, and 68.42% had no lymph node metastasis (N0). Tumors were 73.68% HR+HER2- and 26.32% HR+HER2+, with 96.05% ER-positive and 78.95% with ≥20% PR. Most tumors were grade 2 (56.58%), followed by grade 3 (34.21%). The control group had no chronic diseases, infections, or recent antibiotic use.

### 2.2. Differences in Fecal Microbiota Richness and Diversity Between Breast Cancer Patients and Healthy Controls

At the species level, different alpha diversity indices, including the Shannon (Figure 1a), Chao1 (Figure 1b), and ACE (Figure 1c) indices, were calculated to evaluate the alpha diversity within the fecal microbiota of the study groups. The only index that exhibited significant differences among our three study groups was the ACE index (Shannon *p* = 0.77; Chao 1 *p* = 0.12; ACE *p* = 0.016). The ACE index values for each group demonstrated a noteworthy reduction in diversity in the HR+HER2+ subgroup compared to the control group (*p* = 0.012) and the HR+HER2- subgroup (*p* = 0.013). However, no statistically significant differences were observed between the control and HR+HER2- groups (*p* = 0.438).

Furthermore, beta diversity, reflecting dissimilarities in microbiota communities among the study groups, was evaluated using the Bray–Curtis dissimilarity index. Principal component analysis plots revealed a notable separation in bacterial communities among the control, HR+HER2-, and HR+HER2+ groups (PERMANOVA, *p* < 0.044) (Figure 2).

### 2.3. Variations in the Taxonomic Composition of the Fecal Microbiota Between Study Groups

The analysis of the fecal microbiota at the phylum level indicated that Bacteroides, Firmicutes, Proteobacteria, and Actinobacteria were the predominant phyla, with significant differences observed among all three study groups. Comparing the abundance of these phyla between the study groups, there was a notable increase in Actinobacteria in the control group compared to that in the HR+HER2- BC patients (q < 0.01), as well as increased levels of Bacteroidetes in the control subjects compared to those in the HR+HER2+ BC group (*p* < 0.01). Conversely, the HR+HER2- group displayed a greater abundance of Bacteroidetes (*p* < 0.01), Firmicutes (*p* < 0.01), and Proteobacteria (*p* < 0.01) than the HR+HER2+ group (Figure 3).

At the genus level, significant differences in microbial composition were observed among the study groups. In the control group, we noted a significant increase in the abundance of *Bifidobacterium* (q < 0.01) compared to that in the HR+HER2- group and in the abundance of *Bacteroides* (q < 0.05), *Blautia* (q < 0.05), and *Lawsonia* (q < 0.05) compared to that in the HR+HER2+ group. When comparing the two BC groups, there was a significant increase in *Bacteroides* (q < 0.01), *Blautia* (q < 0.05), *Lachnospira* (q < 0.05), *Lawsonia* (q < 0.01), and *Oscillospira* (q < 0.01) in the HR+HER2- group compared to the HR+HER2+ group (Figure 4).

Finally, at the species level, significant increases in the abundances of *Bifidobacterium adolescentis* (control vs. HR+HER2-, q < 0.05; control vs. HR+HER2+, q < 0.05) and *Bifidobacterium longum* (control vs. HR+HER2-, q < 0.01; control vs. HR+HER2+, q < 0.05) were detected in the control group compared to the BC group. The abundances of *Bacteroides eggerthii* (HR+HER2- vs. control, q < 0.01; HR+HER2+ vs. control, q < 0.05) and *Bacteroides caccae* (HR+HER2+ vs. control, q < 0.04; HR+HER2- vs. control, q < 0.05) were significantly greater in BC patients than in controls. Finally, *Dorea formicigenerans* was increased in HR+HER- patients compared to HR+HER2+ patients (q < 0.01) (Figure 5).

### 2.4. Analysis of the Melatonergic Pathway in Breast Cancer Patients and Healthy Controls

Figure 6a shows significantly greater serum serotonin levels in BC patients than in controls (means ± SDs: 109.83 ng/mL ± 44.34) (*p* < 0.01 in the case of HR+HER2- and *p* < 0.05 in the case of HR+HER2+). However, there were no significant differences in serotonin levels between the HR+HER2- (mean ± SD: 164.11 ng/mL ± 64.52) and HR+HER2+ (mean ± SD: 145.61 ng/mL ± 31.24) groups. Interestingly, a similar trend was observed for the NAS, which was significantly greater in both the HR+HER2- group (52.02 pg/mL ± 21.07) and the HR+HER2+ group (53.17 pg/mL ± 13.09) (*p* < 0.01) than in the control group (32.99 pg/mL ± 12.58) (Figure 6b). Furthermore, serum melatonin levels were significantly lower in individuals with BC than in controls (10.2 pg/mL ± 3.14) (*p* < 0.001). Specifically, HR+HER2+ patients exhibited the lowest level of this hormone (4.05 pg/mL ± 2.51), followed by HR+HER2- patients (6.27 pg/mL ± 2.14) (Figure 6c). Notably, significant differences were observed not only between the two groups and the controls but also between the two groups (*p* < 0.05).

When comparing the NAS/melatonin ratio between the controls and BC patients, it became apparent that this ratio was significantly greater in BC patients. Furthermore, when examining variations in the NAS/melatonin ratio across different subgroups, significant differences were noted between the three study groups (*p* < 0.001), notably with a higher ratio observed in HR+HER2+ patients (11.76) followed by HR+HER2- patients (8.42) (Figure 7).

As depicted in Figure 8a, there was an increase in serum AANAT protein levels (responsible for converting serotonin into NAS) in HR+HER2- (means ± SDs: 8.59 ng/mL ± 3.05) and HR+HER2+ (means ± SDs: 7.34 ng/mL ± 1.39) BC patients compared to controls (means ± SDs: 4.96 ng/mL ± 1.94) (*p* < 0.001 and *p* < 0.05, respectively). However, there were no significant differences in AANAT levels between the BC subgroups. In contrast, a different trend was observed for ASMT protein levels (responsible for converting NAS into melatonin), which were significantly lower in HR+HER2- (3.04 ng/mL ± 1.05) and HR+HER2+ (2.74 ng/mL ± 1.30) BC patients than in controls (4.46 ng/mL ± 1.52) (*p* < 0.01) (Figure 8b). Finally, the analysis of 14-3-3 protein levels revealed no significant differences among any of the groups (controls: 2.65 ng/mL ± 0.80; HR+HER2-: 2.94 ng/mL ± 1.03; HR+HER2+: 2.72 ng/mL ± 0.82, *p* >0.05) (Figure 8c).

### 2.5. Elevated Serum Estrogen Levels and Fecal Beta-Glucuronidase Activity in Breast Cancer Patients Compared to Healthy Controls

Significantly elevated serum estrogen levels were observed in both the HR+HER2- and HR+HER2+ groups compared to the control group (*p* < 0.01). However, no significant differences were identified between the HR+HER2- and HR+HER2+ groups (Figure 9a). Similarly, fecal βGD activity was significantly greater in both the HR+HER2- and HR+HER2+ groups than in the control group (*p* < 0.01), but no differences were found between the BC groups (Figure 9b).

### 2.6. Differences in the Fecal Levels of Short-Chain Fatty Acids (SCFAs) and Serum Zonulin Levels Between Study Groups

A significant increase in acetic acid (AA) was observed in the HR+HER2+ subgroup compared to the control group (*p* < 0.05) and HR+HER2- group (*p* < 0.01). Similarly, isobutyric acid (IBA) exhibited a similar trend, with a significant increase in the HR+HER2+ and HR+HER2- groups compared to the control group (*p* < 0.001 and *p* < 0.05, respectively). No significant differences were found in the fecal levels of propionic acid (PA) (*p* = 0.17), butyric acid (BA) (*p* = 0.59), or valeric acid (VA) (*p* = 0.40) between the study groups. Finally, isovaleric acid (IVA) levels in the HR+HER2+ group were significantly greater than those in the control group (*p* < 0.05) (Figure 10a). In addition, serum zonulin levels were significantly lower in the control group (2.19 ng/mL ± 1.45) than in the other two groups of BC patients: HR+HER2- (5.41 ng/mL ± 4.16, *p* < 0.01) and HR+HER2+ (6.06 ng/mL ± 4.30, *p* < 0.01) (Figure 10b).

### 2.7. Associations Between the Fecal Microbiota and the Serotonin–NAS–Melatonin Axis and Fecal Beta-Glucuronidase Activity

Correlation analyses were conducted to investigate relationships between different bacterial species exhibiting significant differences in abundance between the control group and hormonal receptor-positive BC patients (comprising both HER2- and HER2+ subtypes) and the primary components of the serotonin–NAS–melatonin axis. Additionally, correlations were explored with the activity of βGD in feces to establish possible connections. The serum levels of melatonin were significantly positively correlated with *Faecalibacterium prausnitzii* (*p* = 0.008, r = 0.278), *Ruminococcus bromii* (*p* = 0.046, r = 0.210), and *Bifidobacterium longum* (*p* = 0.030, r = 0.228) (Figure 11). In addition, there was a significant negative correlation between NAS and *Collinsella aerofaciens* (*p* = 0.026, r = −0.233). Moreover, the serum levels of serotonin were significantly negatively correlated with *Collinsella aerofaciens* (*p* = 0.044, r = −0.213), and *Bacteroides uniformis* (*p* = 0.006, r = −0.287). Finally, fecal βGD activity correlated positively with *Bacteroides eggerthii* (*p* = 0.029, r = 0.268), and negatively with *Bifidobacterium longum* (*p* = 0.001, r = −0.384) (Figure 11).

### 2.8. Serum Levels of Serotonin, N-Acetylserotonin, and Melatonin as Biomarkers for Breast Cancer Risk

The serum levels of serotonin, NAS, and melatonin were measured in BC patients and controls, and receiver operating characteristic (ROC) curve analysis was conducted to determine their discriminatory power. Serum serotonin and melatonin demonstrated significant discriminatory capacity for distinguishing BC patients from controls, with area under the curve (AUC) values of 0.855 (95% CI: 0.698–1.012) and 0.838 (95% CI: 0.700–0.976), respectively. NAS exhibited the highest discriminatory power among the three biomarkers, with an AUC of 0.879 (95% CI: 0.754–1.004). Additionally, a composite score incorporating serotonin, the NAS, and melatonin was calculated, and ROC curve analysis was performed to evaluate its diagnostic performance. Combining serotonin, NAS, and melatonin improved diagnostic performance, resulting in a greater AUC of 0.895 (95% CI: 0.765–1.026) than that of the individual biomarkers. Furthermore, the NAS/melatonin ratio showed robust discriminatory ability (AUC: 0.995, 95% CI: 0.980–1.011) for distinguishing hormone receptor-positive BC patients from controls (Figure 12).

## 3. Discussion

This study identified alterations in gut microbiota composition, microbial metabolic activity, melatonergic pathway, and intestinal permeability in hormone receptor-positive (HR+) breast cancer (BC) patients compared to healthy controls. The key finding was a significant increase in the NAS/melatonin ratio in BC patients, accompanied by gut dysbiosis, elevated fecal β-glucuronidase activity and SCFAs levels, and higher serum zonulin levels.

Our results revealed that HR+ BC patients exhibited gut dysbiosis associated with increased serum levels of serotonin and NAS, elevated fecal βGD activity, and reduced serum melatonin levels compared to healthy controls. Additionally, BC patients showed higher fecal levels of SCFAs, such as IBA and IVA, as well as increased serum zonulin. Elevated zonulin levels have previously been associated with intestinal dysbiosis and increased intestinal permeability [18], which facilitate the translocation of gut bacteria and compounds such as lipopolysaccharides, promoting inflammation and interacting with the immune system to enhance cancer susceptibility [19].

A bidirectional relationship exists between the gut microbiota, melatonin production and BC. Alterations in the intestinal microbiota influence the regulation of the melatonergic pathway, while changes in melatonin synthesis affect the microbiota composition [20]. Both factors may contribute to HR+ BC development. In our study, both BC groups exhibited decreased gut microbiota diversity compared to controls. Furthermore, Bray–Curtis PCoA analysis revealed that the microbiota profiles of BC patients clustered separately from those of healthy controls, indicating significant disease-associated microbial shifts.

At the taxonomic level, we observed significant differences between groups. The enrichment of *Bacteroides eggerthii* and *Bacteroides caccae* were detected in HR+HER2+ and HR+HER2- BC patients, while *Bifidobacterium adolescentis* and *Bifidobacterium longum* were enriched in healthy controls. These results were consistent with previous findings in colorectal cancer, where increased *Bacteroides eggerthii* [21] and decreased *Bifidobacterium adolescentis* abundance were reported [22]. Notably, *Bifidobacterium longum*, isolated from breast milk and infant feces, has demonstrated anti-proliferative and antimicrobial activities in the animal models of BC, suggesting a potential protective role in both BC [23] and colorectal cancer [24].

In addition, our study revealed elevated serum serotonin and NAS levels and decreased melatonin concentrations in HR+ BC patients. These findings aligned with prior studies demonstrating that human BC cells (e.g., MCF-7, Bcap-37) and mammary epithelial cells (MCF-10A) can synthesize serotonin and melatonin, with higher serotonin expression observed in BC tissues compared to non-tumorous tissue [25,26]. Dysregulation of serotonin signaling—including receptor expression—has been implicated in BC pathogenesis by affecting cell cycle regulation, autophagy, and apoptosis, thereby promoting proliferation and apoptosis resistance [27,28]. Moreover, serotonin receptor expression has been associated with estrogen receptor and HER2 expression, suggesting that serotonin plays a role in BC progression [29].

Our study provides the first report of elevated NAS levels in HR+ BC patients. NAS plays distinct biological roles, including activation of the TrkB receptor via dimerization and autophosphorylation, triggering pathways involved in BC cell proliferation and metastasis [9,30]. In contrast, decreased melatonin levels in BC patients have been consistently reported [31,32,33]. Melatonin exerts anti-estrogenic effects by modulating aromatase activity and regulating estrogen-metabolizing enzymes such as estrogen sulfotransferase (EST) and sulfatase (STS), which results in higher levels of sulfoconjugated (inactive) estrogens excreted in bile and reduced levels of biologically active estrogens [34].

We also found a significant difference in melatonin levels between the two BC subgroups, with higher levels in the HR+HER2- group compared to the HR+HER2+ group. Although this specific observation has not been previously reported, several studies have linked melatonin with HER2+ BC. For instance, melatonin supplementation has been shown to inhibit mammary tumor development and downregulate HER2/neu oncogene expression in transgenic mice [35]. Another study demonstrated that melatonin suppresses metastasis in HER2+ BC cells via RSK2 inhibition [36].

Moreover, we detected alterations in the protein levels of AANAT and ASMT—key enzymes in the melatonergic pathway. AANAT, which catalyzes the conversion of serotonin to NAS, was elevated, whereas ASMT, which converts NAS to melatonin, was reduced in BC patients. Tran et al. [37] reported that higher ASMT levels correlated with better relapse-free and metastasis-free survival, particularly after tamoxifen therapy. Although no direct association between elevated AANAT and BC has been previously described, SNPs in AANAT regulatory regions (rs4238989, rs3760138) have been linked to increased BC risk [38]. These results suggested that melatonin biosynthesis in HR+ BC may be disrupted due to imbalances in AANAT and ASMT levels, favoring NAS accumulation over melatonin production.

SCFAs produced by gut microbiota may indirectly influence melatonin production by stimulating serotonin synthesis. SCFAs interact with enteroendocrine receptors to induce intestinal hormone secretion, including serotonin, which can affect tumor proliferation and apoptosis [26,39]. Our data showed significantly elevated fecal IVA levels in HR+HER2+ patients. IVA has been reported to promote serotonin synthesis by upregulating tryptophan hydroxylase 2 (Tph2), the rate-limiting enzyme in serotonin production [40]. Moreover, IBA and IVA also modulate epithelial function and tight junctions via G-protein coupled receptors (e.g., GPR41, GPR43). Simultaneously, increased βGD activity promotes estrogen deconjugation and reabsorption, elevating systemic estrogen levels. Both mechanisms may contribute to increased zonulin expression, which disrupts tight junctions and promotes intestinal permeability. This “leaky gut” allows for the translocation of microbial components (e.g., LPS), fostering a pro-tumorigenic environment.

We further analyzed the associations between gut microbial taxa and the serotonin–NAS–melatonin axis, as well as feral βGD activity. A positive correlation emerged between *Bifidobacterium longum* and serum melatonin levels. Previous studies in ulcerative colitis showed that melatonin supplementation increased *Bifidobacterium* abundance while reducing pathogenic genera such as *Desulfovibrio*, *Peptococcaceae*, and *Lachnospiraceae* [41]. In our study, *Bifidobacterium longum* negatively correlated with fecal βGD activity, consistent with findings showing decreased βGD activity after fermented milk consumption enriched with *Bifidobacterium* spp. (e.g., *B. longum* SPM1207) [42]. Conversely, *Bacteroides eggerthii* positively correlated with βGD activity, in line with its prominent role in encoding βGD enzymes—accounting for over 50% of all βGDs identified in the human gut [43].

Dysbiosis in BC patients, characterized by low *Bifidobacterium longum* and high *Bacteroides eggerthii*, may enhance βGD-mediated estrogen reactivation, increasing BC risk. Though our clinical findings are correlative, functional evidence from animal models supports causality. For example, species from *Bifidobacterium* spp. administration in tumor cell lines and murine models inhibit tumor growth and modulate immunity in colon and breast cancer [44,45,46], whereas *Bacteroides eggerthii* promotes colitis in mice, which is related with inflammatory pathways and intestinal permeability [47,48]. Nonetheless, causality must be confirmed through direct microbial manipulation in BC models.

Finally, the NAS/melatonin ratio displayed strong discriminatory power between BC patients and controls. Disruptions in this ratio have been described in conditions like endometriosis, autism, glioblastoma, and depression [17,49,50,51,52,53,54]. The elevated NAS/melatonin ratio observed in BC may result from dysregulated enzyme activity—elevated AANAT and reduced ASMT—favoring NAS accumulation. This metabolic shift may be driven by tumor-related inflammation, microbiota-derived metabolites (e.g., SCFAs), or estrogen feedback. Notably, NAS itself may promote tumor progression through TrkB activation, compounding the dysregulation.

However, this study had limitations. First, the sample size was relatively small, and the single-center design may have introduced selection bias. Second, its cross-sectional nature precluded causal inferences. Additionally, melatonin was measured only in morning serum samples, despite its known circadian rhythm [55]. Nevertheless, since samples from all participants were collected at the same time, relative comparisons remained valid and highlighted reduced melatonin levels in BC patients.

Strengths of this study included its rigorous design, strict inclusion/exclusion criteria, and well-matched control and patient cohorts. Collectively, our findings underscore a complex interplay among gut microbiota, microbial metabolites, intestinal permeability, and the melatonergic pathway in HR+ BC. While causal relationships cannot be established, the identified associations—particularly the NAS/melatonin imbalance and increased βGD activity—merit further mechanistic investigation. The NAS/melatonin ratio, although promising, should be validated in larger, multicentric cohorts before being considered a reliable biomarker.

## 4. Materials and Methods

### 4.1. Study Design and Participants

A cross-sectional study was performed on 76 female patients diagnosed with primary estrogen receptor-positive, stage I-III BC (20 HR+HER2+ and 56 HR+HER2-) and 16 healthy women aged 44–85 years. All subjects were recruited from the Medical Oncology Service at the Virgen de la Victoria University Hospital of Málaga. The diagnosis of hormone receptor-positive BC was established based on expert clinical assessments by pathologists. Information about clinicopathological data and patient follow-up, including demographic details such as age and specific tumor characteristics like grade, size, lymph node involvement, ER/PR expression, and Her2 status, were sourced from pathology reports.

The exclusion criteria were as follows: triple negative BC; other cancer types or a prior history of BC; synchronous tumors; metastatic disease; breast infection within 3 months prior to study inclusion; pregnancy, lactation, autoimmune, inflammatory, or gastrointestinal diseases; the use of exogenous melatonin or psychotropic drugs; and the use of antibiotics, probiotics, and/or prebiotics within the preceding 3 months before the study.

Both patients and controls were instructed to abstain from consuming foods rich in tryptophan and/or serotonin (e.g., chocolate, nuts, coffee, tea, and bananas). They were also advised to limit their intake of caffeine, alcohol, and nicotine particularly close to bedtime. Additionally, participants were instructed to avoid eating and physical activity later in the day and to reduce exposure to artificial light at night for 15 days preceding blood collection. No significant differences in adherence to these recommendations were observed between patients and controls, as evaluated using a structured questionnaire. Blood and fecal samples were collected in the morning between 8:00 a.m. and 10:00 a.m. at baseline before any treatment commenced.

This study obtained approval from the Medical Ethics Committee of the Virgen de la Victoria University Hospital of Málaga and adhered to Good Clinical Practice and the Declaration of Helsinki. Written informed consent was obtained from all participants before protocol initiation.

### 4.2. Fecal Sample Processing, DNA Extraction, and Gut Microbiota Sequencing

DNA was extracted from 200 mg of fecal samples using the QIAamp Fast DNA Stool Mini Kit following the manufacturer’s protocol (Qiagen, Hilden, Germany). DNA concentration and purity were measured with the Qubit 2.0 Fluorometer using the dsDNA Broad Range (BR) Assay Kit (Thermo Fisher Scientific, Waltham, MA, USA). The ribosomal 16S rRNA gene was amplified from the extracted DNA using the Ion 16S Metagenomics Kit (Thermo Fisher Scientific, Madrid, Spain), which contains two primer sets targeting specific hypervariable regions of the 16S rRNA gene in bacteria: primer set V2–4–8 and primer set V3–6, 7–9. Library preparation was completed with the Ion Plus Fragment Library Kit (Thermo Fisher Scientific, Madrid, Spain), and the barcoding of each sample was performed using the Ion Xpress Barcode Adapters kit (Thermo Fisher Scientific, Madrid, Spain). Emulsion PCR and sequencing of the amplicon libraries were carried out on an Ion 530 chip using the Ion Torrent S5™ system and the Ion 510/520TM/530TM Kit-Chef (Thermo Fisher Scientific, Madrid, Spain) according to the provided instructions. To ensure sequencing data quality, artificial mock communities with known composition were included as positive controls. Negative controls were also added to each library to monitor potential contaminants, using DNA extraction blanks and no-template amplification controls.

### 4.3. Bioinformatics Analysis

Base calling and run demultiplexing were carried out using the Torrent Suite™ Server software (Thermo Fisher Scientific), version 5.4.0, with default settings for targeted 16S sequencing (bead loading ≤30, key signal ≤30, and usable sequences ≤30). Quality sequences were processed into amplicon sequence variants (ASVs) using DADA2, with parameters adapted for Ion Torrent data [56], within the open-source software Quantitative Insights into Microbial Ecology (QIIME2, version 2023.7) [57]. This software was also used for diversity and taxonomic analyses. Alpha diversity was evaluated using indexes such as Shannon, ACE, and Chao1, with significance tested via the Kruskal–Wallis test. Beta diversity was calculated using the Bray–Curtis dissimilarity index at species level, and differences between group compositions were assessed through permutational multivariate analysis of variance (PERMANOVA). Taxonomic classification was performed by clustering sequences with VSEARCH, using the Greengenes reference database (version 13_8) at 97% sequence identity. Differential bacterial abundance was analyzed using the DESeq2 (v1.22.2) Bioconductor package [58], with multiple testing correction applied using the Benjamini–Hochberg method, considering a q-value of less than 0.05 as significant.

### 4.4. Analysis of Serum Serotonin, N-Acetylserotonin, and Melatonin Levels by High-Performance Liquid Chromatography with Tandem Mass Spectrometry (HPLC–MS/MS)

The serum levels of serotonin, NAS, and melatonin were quantified using HPLC–MS/MS at Chemical Sanitary Consulting (CQS Lab). Serotonin and melatonin standards were procured from Merck Life Science, while the NAS standard and deuterated internal standards were obtained from Cayman Chemical. Methanol and water were purchased from Supelco and VWR-BDH Chemicals, respectively, and ethyl acetate and formic acid were obtained from Merck Life Science. All reagents used were of high purity and compatible with HPLC–MS analysis.

For sample preparation, 10 μL of serum was diluted (1:20) with H_2_O:MeOH and spiked with a deuterated internal standard for serotonin. For NAS and melatonin determination, 390 μL of serum was spiked with a deuterated internal standard and subjected to double extraction with ethyl acetate (1 mL). After centrifugation, the organic phase was dried using nitrogen gas. The dry extract was reconstituted with 190 μL of H_2_O:MeOH.

Chromatographic separation was performed on a Phenomenex 00b-4251-B0 Luna C18 column with Solvents A (H_2_O with formic acid) and B (MeOH with formic acid) as the mobile phase. Analyses were conducted in positive ionization mode using a triple quadrupole liquid chromatography–mass spectrometry system (Model 6460; Agilent Technologies, Santa Clara, CA, USA). The Agilent MassHunter Workstation software version B.06.00 controlled the HPLC–MS system, while the Agilent MassHunter Quantitative Analysis software version B.06.00 was used for peak integration and quantitative calculations. The calibration curves spanned the ng/mL range for serotonin and the pg/mL range for melatonin and NAS.

### 4.5. Determination of Serum Protein Levels of AANAT, ASMT, and 14-3-3 Involved in Melatonin Synthesis by Enzyme-Linked Immunosorbent Assay (ELISA)

The serum protein levels of AANAT, ASMT, and 14-3-3 were quantified using the Human AANAT ELISA Kit, Human ASMT ELISA Kit, and Human 14-3-3 Protein (14-3-3 Pro) ELISA Kit, respectively, which were obtained from MyBioSource, Inc. (San Diego, CA, USA), following the manufacturer’s instructions. The mean values were utilized for data analysis. The intra- and inter-assay coefficients of variation were consistently less than 15% across all measurements. The detection limits were 0.094 ng/mL for AANAT, 0.1 ng/mL for ASMT, and 0.5 ng/mL for 14-3-3 Pro [59].

### 4.6. Measurement of Fecal Beta-Glucuronidase Activity

The quantitative determination of βGD activity in fecal samples was performed using the β-Glucuronidase Activity Assay Kit (Fluorometric) (BioVision, Inc., Abcam, Cambridge, MA, USA), with an adaptation of the protocol for fecal samples based on previous studies [60,61]. Fluorescence was measured at an excitation wavelength of 330 nm and an emission wavelength of 450 nm using an FLS920 fluorimeter (Edinburgh Instruments). One unit represented the amount of βGD capable of cleaving 1 μmol of substrate per minute under the assay conditions.

### 4.7. Intestinal Permeability Analysis

The serum levels of zonulin were evaluated in duplicate using the commercial enzyme-linked immunosorbent assay Human Zonulin ELISA Kit 96T (Cusabio, Wuhan, China). The mean values were utilized for data analysis. The intra- and inter-assay coefficients of variation ranged from 3% to 10%, with a detection limit of 0.22 ng/mL.

### 4.8. Analysis of Short-Chain Fatty Acids (SCFAs) in Fecal Samples Was Conducted Using Gas Chromatography Coupled to Triple Quadrupole Mass Spectrometry (GC–MS/MS)

The fecal concentrations of SCFAs were determined using GC–MS/MS following previously described procedures [62] at the Eurecat Centre Tecnològic de Catalunya within the technological infrastructure of the Center for Omic Sciences (COS). Briefly, 20 mg of fecal sample was placed in a 2 mL Eppendorf tube and mixed with 200 μL of internal standard (IS) solution (acetic acid (AA-LAB) at 6000 μM, butyric acid (BA-LAB) at 1200 μM isobutyric acid (IBA-LAB), valeric acid (VA-LAB), isovaleric acid (IVA-LAB), and propionic acid (PA-LAB) at 600 μM) (Merck Life Science, Darmstadt, Germany) along with 200 μL of PBS. The samples were vigorously mixed for 10 min and then centrifuged at 2500 rpm for 10 min at 4 °C. The supernatants (50 μL) were acidified by adding 10 μL of 15% phosphoric acid, followed by SCFA extraction using 1000 μL of MTBE for 10 min. After centrifugation at 1500 rpm for 10 min at 4 °C, the upper organic layer was transferred to a glass vial for analysis. Approximately 50 mg of the sample was placed in a 2 mL Eppendorf tube and subsequently freeze-dried to determine the sample mass on a dry basis.

The SCFAs were separated using a DB-FFAP chromatographic column (30 m × 0.25 mm × 0.25 μm). The temperature in the oven was programmed as follows: (i) initial temperature set at 40 °C, (ii) linear increase at 12 °C/min to 130 °C (0 min), (iii) further linear increase at 30 °C/min to 200 °C (0 min), and (iv) a final step with a temperature ramp at 100 °C/min to 250 °C (4.5 min). The column flow was maintained at 1.5 mL/min using helium as the carrier gas. The injector temperature was set at 250 °C, and the extracts were injected in splitless mode. Ionization was achieved by electronic impact (70 eV), and the mass analyzer was operated in the multiple reaction monitoring (MRM) mode. The *m/z* values and their corresponding retention times are provided in Table 2.

### 4.9. Statistical Analysis

The Kruskal–Wallis rank sum test, followed by the Dunn post hoc test, was utilized to examine differences in serotonin metabolites and associated enzymes, SCFA levels, βGD activity, and zonulin levels among the study groups. Spearman correlation coefficients were computed to estimate correlations between bacterial taxa and the principal components of the serotonin–NAS–melatonin axis, as well as βGD. Statistical analyses were conducted using SPSS software version 26.0 (SPSS Inc., Chicago, IL, USA). The p-values less than 0.05 were considered to indicate statistical significance.

## 5. Conclusions

In conclusion, our findings support the existence of a potential gut microbiota–melatonergic–barrier axis in hormone receptor-positive breast cancer. We observed a disrupted melatonergic pathway, characterized by a significantly increased NAS/melatonin ratio, gut dysbiosis, elevated microbial metabolic activity, and impaired intestinal barrier function. These alterations may contribute to the pathophysiology of BC, although causality cannot be inferred from our data.

The NAS/melatonin ratio emerged as a promising biomarker candidate; however, its diagnostic potential should be interpreted cautiously and requires further validation in larger, independent cohorts, as well as in other BC subtypes. Future research integrating functional and longitudinal approaches will be crucial to elucidate the mechanistic role of gut microbiota in breast cancer development and progression, and to evaluate novel preventive or therapeutic strategies targeting the gut–brain–tumor axis.

## Figures and Tables

**Figure 1 ijms-26-06801-f001:**
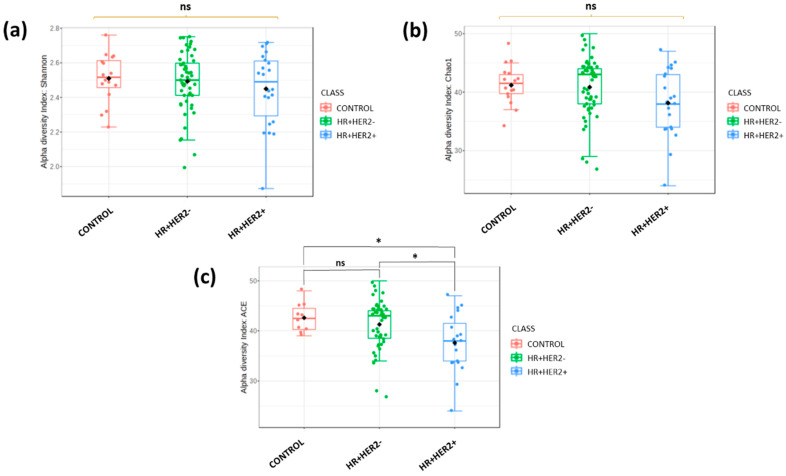
Comparison of alpha diversity among different study groups. (**a**) Shannon index; (**b**) Chao 1 index; (**c**) ACE index. ns: no significance; * *p* < 0.05.

**Figure 2 ijms-26-06801-f002:**
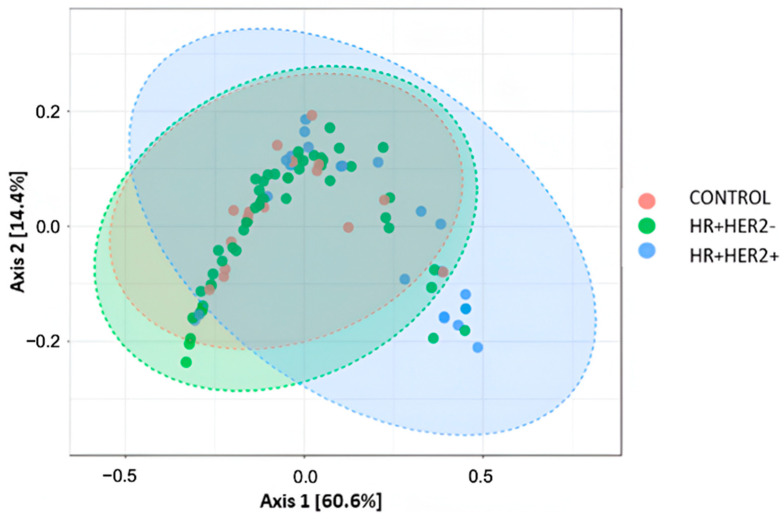
Beta diversity in feces among the study groups analyzed via PCA using Bray–Curtis dissimilarity matrix at the species level. Each data point represents a community labeled according to the study groups. The percentage of variation explained by the principal coordinates is shown on the axes.

**Figure 3 ijms-26-06801-f003:**
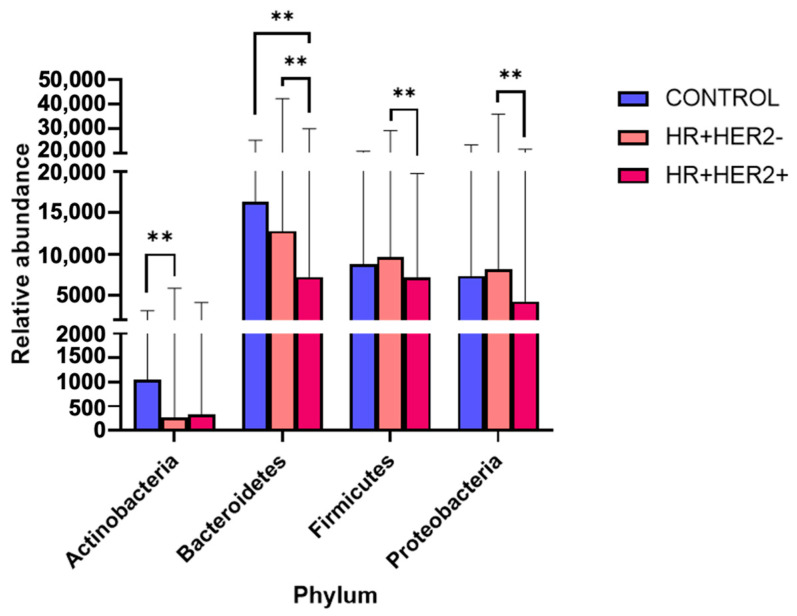
Bacterial phyla in fecal samples from controls and HR+HER2- and HR+HER2+ breast cancer patients. The data are presented as relative abundances. Bars represent the medians and standard deviations. ** *p* < 0.01.

**Figure 4 ijms-26-06801-f004:**
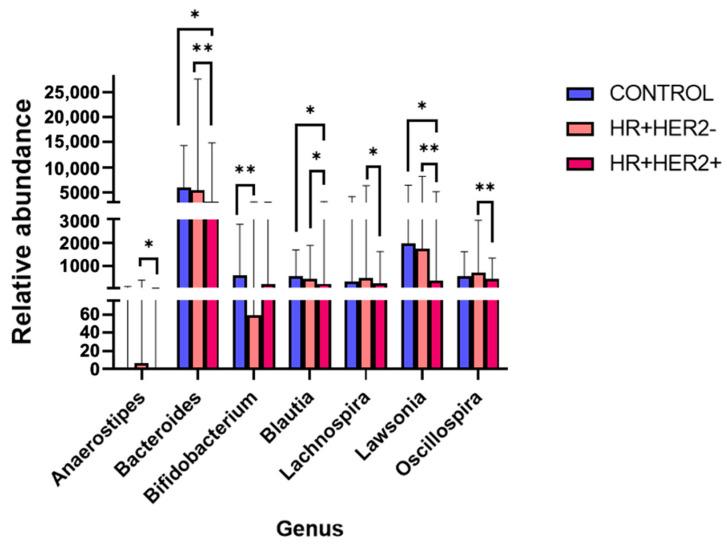
Levels of relative abundance for genera with significant differences between groups. * *p* < 0.05; ** *p* < 0.01.

**Figure 5 ijms-26-06801-f005:**
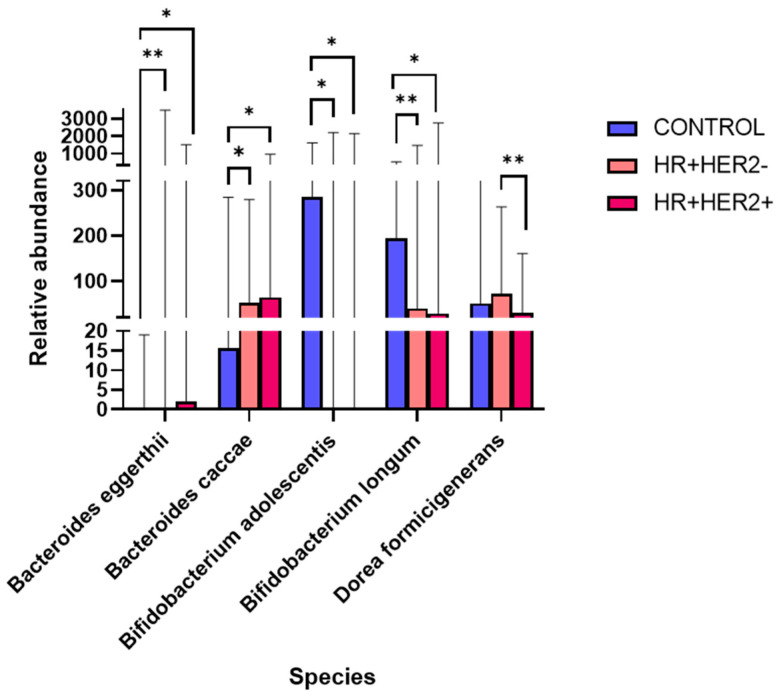
Relative abundance of bacterial species with significant differences between study groups. * *p* < 0.05; ** *p* < 0.01.

**Figure 6 ijms-26-06801-f006:**
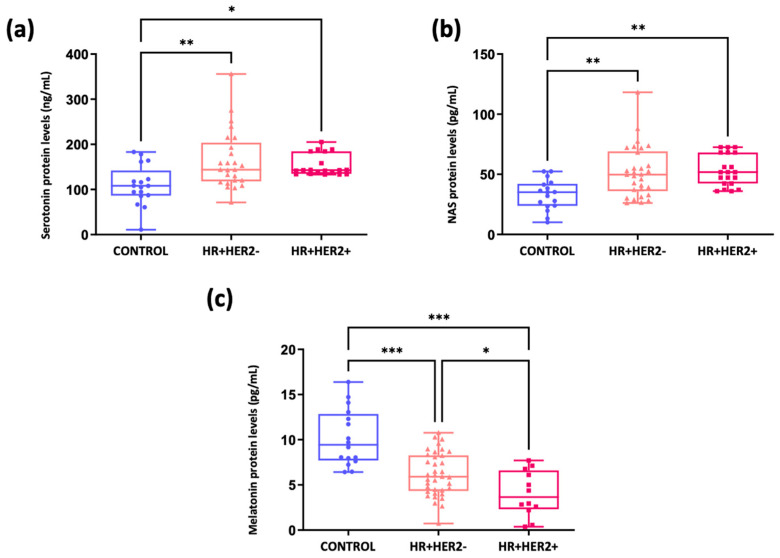
Measurement of serum serotonin, NAS and melatonin levels. (**a**) Serotonin (ng/mL); (**b**) NAS (pg/mL); (**c**) Melatonin (pg/mL) in controls and HR+HER2- and HR+HER2+ breast cancer patients. * *p* < 0.05; ** *p* < 0.01; *** *p* < 0.001.

**Figure 7 ijms-26-06801-f007:**
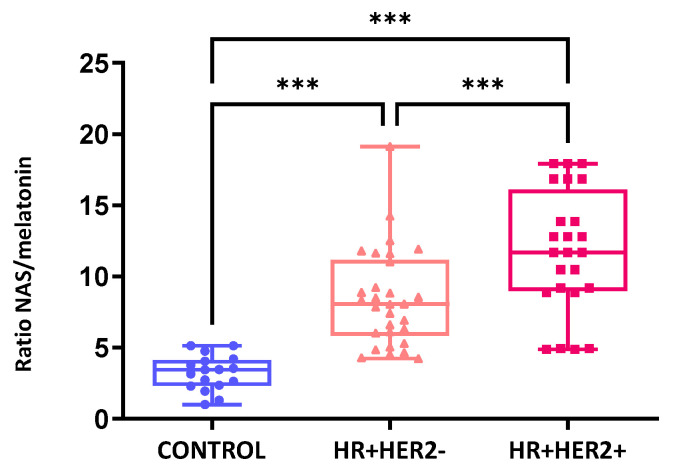
Ratio of NAS to melatonin in the control and breast cancer patient groups. *** *p* < 0.001.

**Figure 8 ijms-26-06801-f008:**
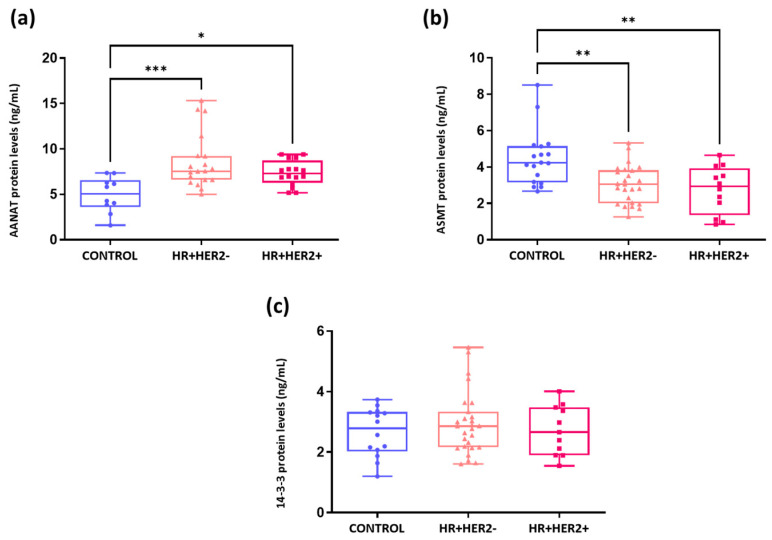
Measurement of serum AANAT, ASMT, and 14-3-3 protein levels in control and HR+HER2- and HR+HER2+ breast cancer patients. (**a**) AANAT (ng/mL); (**b**) ASMT (ng/mL); (**c**) 14-3-3 (ng/mL). * *p* < 0.05; ** *p* < 0.01; *** *p* < 0.001.

**Figure 9 ijms-26-06801-f009:**
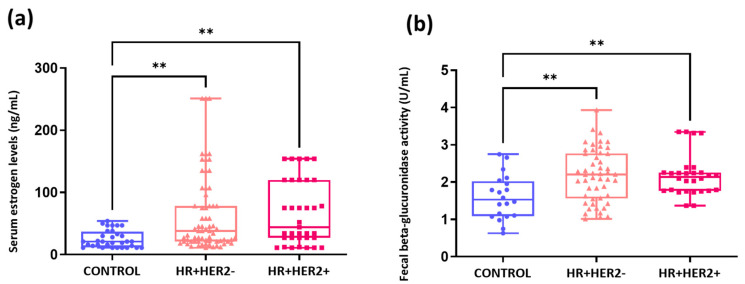
Measurement of serum estrogen levels (ng/mL) (**a**) and beta-glucuronidase activity in feces (nU/mL) (**b**). ** *p* < 0.01.

**Figure 10 ijms-26-06801-f010:**
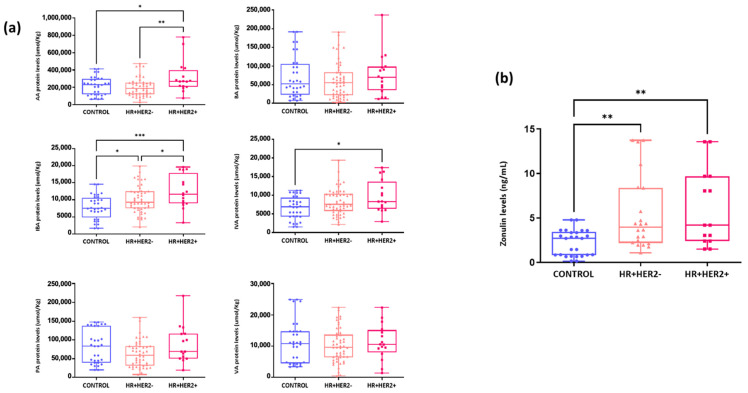
Quantification of fecal short-chain fatty acids: (**a**) acetic acid (AA), butyric acid (BA), isobutyric acid (IBA), isovaleric acid (IVA), propionic acid (PA), and valeric acid (VA) in fecal samples (μmol/kg) and (**b**) serum zonulin levels (ng/mL) among study groups. * *p* < 0.05; ** *p* < 0.01; *** *p* < 0.001.

**Figure 11 ijms-26-06801-f011:**
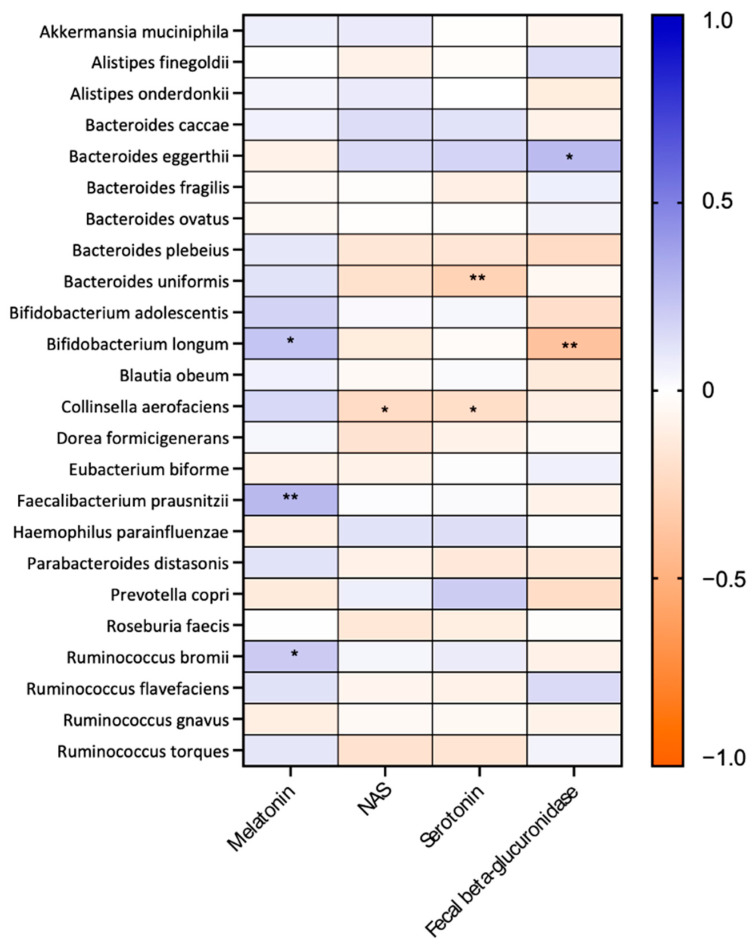
Heatmap analysis conducted to evaluate the relationships between the serum levels of components of the serotonin–NAS–melatonin axis and fecal beta-glucuronidase activity with fecal microbiota in hormonal receptor-positive BC patients and controls. * *p* < 0.05; ** *p* < 0.01.

**Figure 12 ijms-26-06801-f012:**
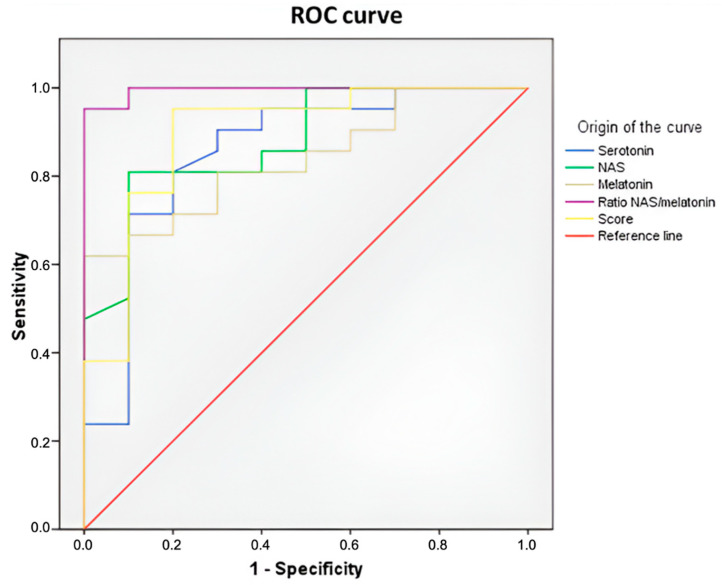
ROC curve analysis and AUC values were employed to evaluate the diagnostic potential of the serum levels of serotonin, NAS, melatonin; the combined score; and the NAS/melatonin ratio as biomarkers for hormone receptor-positive breast cancer. AUC: area under the curve; ROC: receiver operating characteristic.

**Table 1 ijms-26-06801-t001:** Clinicopathological characteristics of the study subjects.

	Controls n = 16	BC Patients n = 76
Age (Years) Average (range)	62 (47–85)	64 (44–85)
HORMONAL STATUS		
Perimenopausal	2 (12.5%)	9 (11.84%)
Postmenopausal	14 (87.5%)	67 (88.15%)
ESTROGEN STATUS	Not applicable	
Positive		73 (96.05%)
Negative		3 (3.95%)
PROGESTERONE STATUS	Not applicable	
≥20%		60 (78.95%)
<20%		7 (9.21%)
Negative		9 (11.84%)
HER2 STATUS	Not applicable	
Positive		20 (26.32%)
Negative		56 (73.68%)
SUBTYPE CLASSIFICATION	Not applicable	
HR+HER2-		56 (73.68%)
HR+HER2+		20 (26.32%)
AFFECTED AXILAR LYMPH NODE	Not applicable	
N0		52 (68.42%)
N1		17 (22.37%)
N2		7 (9.21%)
TUMOR SIZE (cm)	Not applicable	
<2		32 (42.11%)
2 to 5		35 (46.05%)
>5		9 (11.84%)
TUMOR TYPE	Not applicable	
IDC (invasive ductal carcinoma)		61 (80.26%)
ILC (invasive lobular carcinoma)		11 (14.47%)
Papillary carcinoma		2 (2.63%)
Mucinous carcinoma		2 (2.64%)
HISTOLOGICAL GRADE	Not applicable	
Grade 1		7 (9.21%)
Grade 2		43 (56.58%)
Grade 3		26 (34.21%)

**Table 2 ijms-26-06801-t002:** Details of the optimized MRM transition for each analyte.

Analyte	Retention Time (min)	Quantitative Transition (*m/z*)	Qualitative Transition (*m/z*)	CE (V)
AA	6.62	60→45	60→43	10/5
PA	7.49	73→55	74→55	5/10
IBA	7.77	88→73	88→55	10/15
BA	8.27	60→42	73→55	10/5
IVA	8.56	60→42	60→45	10/10
VA	9.00	60→42	73→55	10/5

## Data Availability

The 16S rRNA metagenomic sequences for the study cohorts are available via SRA (SUB14552390) with BioProject number PRJNA1127492.

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
