# Peer review of "Exploring the Interplay Between Gut Microbiota and the Melatonergic Pathway in Hormone Receptor-Positive Breast Cancer"

_ijms, 2025, doi:10.3390/ijms26146801_

Round 1

Reviewer 1 Report

Comments and Suggestions for Authors

The present study explores a novel link between the gut microbiota and the melatonergic pathway in HR+ breast cancer. The findings reveal microbial alterations and dysregulation in the serotonin-NAS-melatonin axis, highlighting a potential involvement of the gut-brain-hormone axis in pathophysiology of breast cancer. Its clinical relevance lies in helping to develop future treatments and prevention strategies, and may also improve early diagnosis and lead to more personalized care in a common type of breast cancer.

Although I will recommend its publication in IJMS, some minor revisions are needed:

  1. Table 2: Clinicopathological characteristics from the control subjects are excluded. Including these data would enhance the interpretability of the results.
  2. Image quality: The quality of the images could be improved.
  3. The role of NAS/melatonin ratio is quite interesting but its causal mechanisms of its alterations in pathological context should be better justify.
  4. Strengthen the connection between β-glucuronidase activity, SCFA levels and intestinal permeability. Although, the association is reported, the causal mechanism is not discussed in depth.
  5. The introduction does not present the specific hypothesis/rationale for combining the melatonergic pathway with microbiota analysis.

Author Response

For research article

Response to Reviewer 1 Comments

1. Summary

We sincerely thank Reviewer 1 for their thorough evaluation of our manuscript and for the constructive feedback provided. We have carefully considered each of the comments and have revised the manuscript accordingly to improve its clarity and scientific rigor. Below, we provide detailed, point-by-point responses to the reviewer’s observations. All changes made are highlighted in the revised manuscript using track changes. We are confident that the revisions have significantly strengthened the quality and clarity of the work.

2. Questions for General Evaluation

Reviewer’s Evaluation

Response and Revisions

Does the introduction provide sufficient background and include all relevant references?

Can be improved

We thank the reviewer for this observation. In response, we carefully revised the Introduction to ensure it provides a clear and comprehensive background. The current version includes:

-       A summary of established and emerging risk factors for hormone receptor-positive breast cancer (HR+ BC), including hormonal exposure, circadian disruption, and gut dysbiosis.

-       A mechanistic explanation of how gut microbiota, particularly through β-glucuronidase activity, may influence estrogen reabsorption and systemic levels.

-       A detailed overview of melatonin’s role in hormonal regulation and its interactions with gut microbiota, supported by relevant references [1–16].

-       A description of the reciprocal relationship between melatonin and microbial composition, SCFA production, and implications for intestinal integrity and inflammation in BC.

-       The identification of a clear knowledge gap in the literature regarding the joint analysis of microbiota composition, melatonergic pathway metabolites, βGD activity, SCFAs, and intestinal permeability markers in HR+ BC.

We have ensured that all included references are up to date and relevant to the study objectives. If the reviewer had specific references or topics in mind that appear to be missing, we would be happy to consider them for inclusion.

Is the research design appropriate?

Yes

We thank the reviewer for their positive assessment. We are glad that the research design was found to be appropriate for addressing the study objectives.

Are the methods adequately described?

Yes

We thank the reviewer for acknowledging the adequacy of our methodological descriptions. No further revisions were considered necessary in this section.

Are the results clearly presented?

Yes

We appreciate the reviewer’s positive assessment of the results section. We have preserved the structure and clarity of the results, ensuring that all findings remain well-supported by the data and are appropriately interpreted.

Are the conclusions supported by the results?

Yes

We have revised the discussion and conclusion sections to ensure closer alignment with the presented results. Speculative statements have been reduced, and we have emphasized the exploratory nature of our findings. Where relevant, we clarified that certain observed associations warrant further investigation in larger or interventional studies. These changes enhance the coherence between the data and the conclusions drawn.

Are all figures and tables clear and well-presented?

Can be improved

We thank the reviewer for this valuable suggestion. In response, we have made the following improvements:

-       Table 1 (Clinical Characteristics) has been revised and expanded to include more detailed and complete information, thereby improving the contextual understanding of the study population.

-       Figure 2 (Beta diversity plot) and Figure 12 (ROC curve) have been updated with higher-resolution images to ensure better visual clarity and interpretation.

These changes have been implemented in the revised manuscript and are highlighted.

3. Point-by-point response to Comments and Suggestions for Authors

Comments 1: Table 2: Clinicopathological characteristics from the control subjects are excluded. Including these data would enhance the interpretability of the results.

Response 1: The table referenced in the text was actually Table 1, and I have corrected this numbering in the manuscript. In the current Table 1, we have included all available clinical data from both breast cancer patients and healthy controls. For variables that are inherently inapplicable to healthy individuals—such as tumor characteristics, hormone receptor status, or HER2 expression—we have indicated “Not applicable.” This decision reflects the biological and clinical reality that such parameters are only measurable and relevant in patients diagnosed with breast cancer. Including “Not applicable” in these cases avoids misleading assumptions or forced comparisons and allows for a clearer, more accurate presentation of the dataset.

Comments 2: Image quality: The quality of the images could be improved.

Response 2: Thank you for pointing this out. We have replaced the original figures with higher-resolution versions to improve clarity and ensure that all visual details are easily interpretable.

Comments 3: The role of NAS/melatonin ratio is quite interesting but its causal mechanisms of its alterations in pathological context should be better justify.

Response 3: We agree with the reviewer that the mechanistic context of the NAS/melatonin ratio deserves further clarification. In the last three paragraphs of the Discussion (lines 525 to 546), we expanded on this topic. Specifically, we detail how the imbalance between AANAT and ASMT expression in HR+ BC patients may favor NAS accumulation over melatonin synthesis. We also discuss how tumor-related inflammation, gut dysbiosis, and microbial metabolites (e.g., SCFAs) may drive this metabolic shift. Furthermore, we highlight how elevated NAS, via TrkB activation, may contribute to tumor progression. This supports the relevance of the NAS/melatonin ratio as a potentially meaningful biomarker.

Comments 4: Strengthen the connection between β-glucuronidase activity, SCFA levels and intestinal permeability. Although, the association is reported, the causal mechanism is not discussed in depth.

Response 4: Thank you for this insightful comment. We have strengthened this connection in paragraphs 2 (Lines 435 to 442), and especially paragraph 9 (Lines 495 to 506) of the Discussion, where we elaborate on the possible mechanisms linking increased β-glucuronidase activity and SCFA levels (e.g., IBA, IVA) with enhanced intestinal permeability and elevated zonulin levels. We now clarify how SCFAs interact with G-protein coupled receptors to modulate tight junction integrity, and how elevated βGD activity may increase estrogen reabsorption and inflammation, contributing to a pro-tumorigenic environment.

Comments 5: The introduction does not present the specific hypothesis/rationale for combining the melatonergic pathway with microbiota analysis.

Response 5: We appreciate this valuable comment. In response, we have clarified and strengthened the rationale and hypothesis in the last two paragraphs of the Introduction (Lines 76 to 87). Specifically, we now emphasize the bidirectional relationship between melatonin and gut microbiota, including how dysbiosis may alter tryptophan metabolism, reduce SCFA production, and increase the NAS/melatonin ratio—mechanisms potentially involved in hormone receptor-positive breast cancer. Furthermore, we clearly state our hypothesis: that disruptions in the melatonergic pathway and gut microbial metabolism are interrelated processes that contribute synergistically to BC pathogenesis. We also explain the rationale for our integrative analysis, which includes microbial composition, SCFAs, β-glucuronidase activity, melatonin and its precursors, and markers of intestinal permeability.

4. Response to Comments on the Quality of English Language

Point 1: The English is fine and does not require any improvement.

Response 1: We thank the reviewer for the positive feedback regarding the quality of the English language. No changes have been made in this regard.

5. Additional clarifications

Reviewer 2 Report

Comments and Suggestions for Authors

The manuscript reveals gut dysbiosis, alterations in serotonin, NAS, melatonin, βGD activity, and SCFAs in BC pathophysiology. However, the conclusions in this article are obtained from insufficient evidence, and the article is not recommended for publication in the IJMS journal because of the following shortcomings:

  1. The abstract writing does not conform to the journal's standards.
  2. The introduction section has been illogically organized. The authors repeatedly described the significant effects of gut microbiota, melatonin, and breast cancer risk from paragraph 2 to paragraph 4. What’s more, the authors did not summarized current research progress and its absence involving gut microbiota, melatonin, and breast cancer risk.  
  3. NASis suggested to give its full name in its first appearance.
  4. The text of Table 2 has been described in the section 2.1 and can be deleted.
  5. The alterations in gut microbiota composition, etc in BC pathophysiology is measured between BC patients (HR+HER2+ and HR+HER2-) and healthy controls, but they lack the support of other evidence, such as inhibition of Bacteroides eggerthii, and intestinal bacterial inoculation of Bifidobacterium longumin BC patients.
  6. NAS/melatonin ratio showed robust discriminatory ability for distinguishing hormone receptor-positive BC patients from controls in this work, this data can not point to the conclusion that the NAS/melatonin ratio as a biomarker for distinguishing hormone receptor- positive BC patients from healthy individuals due to the lack of other BC populations.

Author Response

For research article

Response to Reviewer 2 Comments

1. Summary

We sincerely thank you very much for your thoughtful and constructive feedback, which has helped us to improve the clarity, structure, and interpretation of our work. We have carefully addressed each comment and revised the manuscript accordingly. Major changes include a reorganization of the introduction for better logical flow, a restructured abstract to meet the journal's format requirements, a more cautious interpretation of the NAS/melatonin ratio, and clearer acknowledgment of the limitations of our microbiota findings. All revisions are highlighted in the re-submitted files, and detailed point-by-point responses are provided below.

2. Questions for General Evaluation

Reviewer’s Evaluation

Response and Revisions

Does the introduction provide sufficient background and include all relevant references?

Must be improved

We thank the reviewer for pointing this out. In response, we have thoroughly revised the introduction to make it more concise and focused. We reduced redundancy in the background information on the microbiota and melatonin pathways, clarified the rationale for combining these elements in the context of breast cancer, and added references to recent and relevant literature. These changes aim to provide a clearer and more compelling justification for our study.

Is the research design appropriate?

Must be improved

We acknowledge the reviewer’s concern. We have now clarified the design in the “Introduction” section, emphasizing the inclusion criteria, justification for the selection of hormone receptor-positive BC patients, and the scope of microbiota and metabolite analyses. Additionally, we added a paragraph in the “Limitations” section to address the lack of interventional or functional analyses and the use of a single cancer subtype. These clarifications aim to better contextualize the design and its interpretative limits.

Are the methods adequately described?

Yes

We thank the reviewer for recognizing the adequacy of our methodological descriptions. No further revisions were needed in this section.

Are the results clearly presented?

Yes

We appreciate the reviewer’s positive evaluation of the results section. We have maintained the structure and clarity of the results, ensuring that all findings are supported by the data and appropriately interpreted.

Are the conclusions supported by the results?

Yes

We have revised the discussion and conclusion sections to better align them with the presented results. We reduced speculative statements and emphasized the exploratory nature of the findings. Where appropriate, we clarified that certain associations observed warrant further investigation in larger or interventional studies. These changes improve the coherence between the data and the conclusions drawn.

Are all figures and tables clear and well-presented?

Yes

We thank the reviewer for this positive comment. Nonetheless, we made minor adjustments to Table 1 for clarity, including the addition of available control data and a “not applicable” (NA) label for clinical variables that do not pertain to the control group, in order to enhance interpretability.

3. Point-by-point response to Comments and Suggestions for Authors

Comments 1: The abstract writing does not conform to the journal's standards.

Response 1: We appreciate this observation and have fully revised the abstract to align with the journal’s structure and formatting guidelines. Specifically, we have ensured clarity in the objectives, methodology, key findings, and conclusions, and avoided any redundancy or ambiguity.

Comments 2: The introduction section has been illogically organized. The authors repeatedly described the significant effects of gut microbiota, melatonin, and breast cancer risk from paragraph 2 to paragraph 4. What’s more, the authors did not summarize current research progress and its absence involving gut microbiota, melatonin, and breast cancer risk.

Response 2: We thank the reviewer for pointing this out. The introduction has been reorganized to follow a more logical progression:

  • First, we contextualize the relevance of breast cancer and hormone receptor subtypes.
  • Then, we introduce the melatonergic pathway and its relevance in carcinogenesis.
  • Afterwards, we present the role of gut microbiota and its metabolic byproducts.
  • Finally, we describe the rationale for combining both lines of research (melatonergic signaling and microbiota) in this study.
    Additionally, we have included a paragraph summarizing the current state of the art and knowledge gaps regarding the link between microbiota, melatonin, and breast cancer, to better justify the novelty of our approach.

Comments 3: NAS is suggested to give its full name in its first appearance.

Response 3: We have corrected this by spelling out the full name—N-acetylserotonin (NAS)—at its first mention in the manuscript, followed by the abbreviation.

Comments 4: The text of Table 2 has been described in the section 2.1 and can be deleted.

Response 4: Thank you for the suggestion. We believe that describing the most relevant data from Table 2 (actually Table 1) in section 2.1 helps highlight the comparability of the groups and supports the interpretation of the results. However, to avoid redundancy, we have condensed the accompanying text and ensured it does not duplicate the table content.

Comments 5: The alterations in gut microbiota composition, etc. in BC pathophysiology is measured between BC patients (HR+HER2+ and HR+HER2-) and healthy controls, but they lack the support of other evidence, such as inhibition of Bacteroides eggerthii, and intestinal bacterial inoculation of Bifidobacterium longum in BC patients.

Response 5: We appreciate this insightful comment. While our study design was observational and did not include functional experiments such as bacterial inhibition or inoculation models, we have now addressed these limitations in the discussion. Specifically, we note that the associations observed in microbiota composition require further experimental validation to confirm causality (Lines 517 to 524). We have also cited relevant studies that support the potential roles of Bacteroides eggerthii and Bifidobacterium longum in breast cancer contexts (References 44 to 47).

Comments 6: “NAS/melatonin ratio showed robust discriminatory ability for distinguishing hormone receptor-positive BC patients from controls in this work, this data cannot point to the conclusion that the NAS/melatonin ratio is a biomarker for distinguishing hormone receptor-positive BC patients from healthy individuals due to the lack of other BC populations.”

Response 6: We agree with this important clarification. The conclusion has been adjusted accordingly: we no longer refer to NAS/melatonin ratio as a definitive biomarker but rather as a potential indicator that warrants further validation in larger and more diverse BC populations, including other subtypes beyond HR+HER2− and HR+HER2+. We have also added this limitation explicitly in the discussion.

4. Response to Comments on the Quality of English Language

Point 1: The English is fine and does not require any improvement.

Response 1: We thank the reviewer for the positive feedback regarding the quality of the English language. No changes have been made in this regard.

5. Additional clarifications

Round 2

Reviewer 2 Report

Comments and Suggestions for Authors

The revised version of this manuscript has improved greatly and given a comprehensive overview in Introductin. The reviewer recommend the article to be published in the IJMS journal.